

# DNA shuffling to improve crude-water interfacial activity in biosurfactants with OmpA protein of *Escherichia coli*

Vanessa Lucía Nuñez Velez[1], Liseth Daniela Villamizar Gomez[1], Jhon E. Mendoza Ospina[1], Yasser Hayek-Orduz[1], Miguel Fernandez-Niño[2], Silvia Restrepo Restrepo[1], Óscar Alberto Álvarez Solano[1], Luis H. Reyes Barrios[1] and Andres F. Gonzalez Barrios[1]

[1] Department of Chemical and Food Engineering, Universidad de los Andes, Bogotá, Cundinamarca, Colombia
[2] Department of Bioorganic Chemistry, Leibniz-Institute of Plant Biochemistry, Hale, Germany

## ABSTRACT

Surfactants are molecules derived primarily from petroleum that can reduce the surface tension at interfaces. Their slow degradation is a characteristic that could cause environmental issues. This and other factors contribute to the allure of biosurfactants today. Progress has been made in this area of research, which aims to satisfy the need for effective surfactants that are not harmful to the environment. In previous studies, we demonstrated the surface tension activity of the *Escherichia coli* transmembrane protein OmpA. Here, we carried out DNA shuffling on *ompA* to improve its interfacial activity. We evaluated changes in interfacial tension when exposing mutants to a water-oil interface to identify the most promising candidates. Two mutants reached an interfacial tension value lower (9.10 mN/m and 4.24 mN/m) than the original protein OmpA (14.98 mN/m). Since predicted isoelectric point values are far from neutral pH, the charge of the protein was a crucial factor in explaining the migration of proteins towards the interface. Low molecular weight mutants did not exhibit a significant difference in their migration to the interface.

## INTRODUCTION

Biosurfactants, such as the rhamnolipids produced by *Pseudomonas aeruginosa* (*P. aeruginosa*), have similar properties to chemical surfactants used in enhanced oil recovery (EOR; *Amirchand & Singh, 2022*), including improved surface tensions, foaming activity, and high-temperature resistance (*Khan & Sasmal, 2022*). They are surface-active compounds primarily synthesized by bacteria, yeasts, and fungi during their cellular processes (*Costa et al., 2018*). Biosurfactants are divided into high molecular and low molecular weight biosurfactants with transmembrane proteins among the with high molecular weight group. These proteins, commonly known as porins, facilitate the transport of nutrients across the cell membrane (*Jiang et al., 2021*).

Corresponding author
Andres F. Gonzalez Barrios,
andresf80@gmail.com

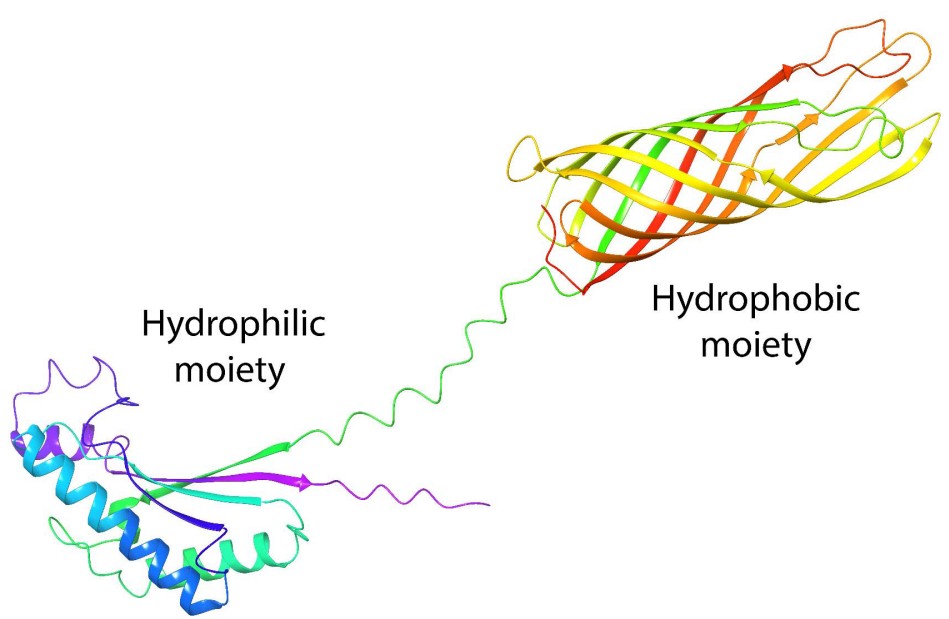

Hydrophilic
moiety

Hydrophobic
moiety

**Figure 1** **Tertiary structure of OmpA predicted by AlphaFold2.**

One of the most studied porins is OmpA from *Escherichia coli* (*E. coli*). Its structure is formed by two domains: the N-terminal (residues 1–170), which spans the cell membrane eight times in anti-parallel beta sheets, and the C-terminal (residues 171–346) (*Aguilera-Segura et al., 2016*), which remains in the periplasm, to control possible interactions with peptidoglycan molecules and regulate osmosis (*Confer & Ayalew, 2013*). These domains in OmpA confer the common properties of a surfactant with well-defined hydrophilic and hydrophobic moieties (Fig. 1), which have been investigated with great interest because they present structural homology with the AlnA protein which forms the surfactant complex Alasan (*Toren et al., 2002*). We previously demonstrated that OmpA facilitates *E. coli* to form biofilms (*Barrios et al., 2006*), with the capability of stabilizing oil-in-water emulsions, and as an alternative to rhamnolipids in EOR processes (*Silva et al., 2014*). In addition, molecular dynamics studies and free energy calculations demonstrate the stability of proteins at the oil-water interface (*Segura et al., 2014b*).

Today's employed chemicals, such as cationic surfactants and polyacrylamides, are toxic to the aquatic and microbial communities, negatively impacting the environment (*Mukhopadhyay, Duttagupta & Mukherjee, 2022*). Interfacial tension (IFT) has been extensively applied to study surface-active chemicals extracted from crude oil (*Bertin et al., 2021*). This property is defined as the effort required to increase the interfacial area between two immiscible phases, as it plays a crucial role in the stability of crude-water emulsion (*DeAlmeida et al., 2016*). Among them, asphaltenes enable the formation of cohesive interfacial films (typical of surfactants with low lateral motility) that generate electrostatic or steric coalescence barriers. They consist of aromatic ring structures with

alicyclic chains that play a significant dispersant role. The acidic regions contribute significantly to lowering IFT (*Jennings et al., 2022*).

The composition and structure of asphaltenes in crude oils vary greatly, necessitating various surfactants capable of competing with them at the interface, thereby generating an unstable environment necessary to separate oil from water. Once the chemical or natural surfactant is injected, it is anticipated that the IFT will decrease, and the magnitude of this decrease is positively correlated with its effectiveness. Oil dispersants and emulsifiers have been proposed to separate (or mix) oil and water. EOR, oil-water demulsification, and chemical flooding are examples of applications.

Due to the complexity of large-scale synthesis and high production costs, the commercial use of biosurfactants is restricted (*Solov'eva et al., 2022*). However, despite these limitations, interest in biosurfactants is clear, as they are non-toxic green molecules that can be used in different industries (bioremediation, pharmaceutical, food, and agriculture) to simplify processes, improve products, facilitate the compliance of safety regulations related to the disposal of industrial wastes (*Carolin et al., 2022*) and to diversify the portfolio of molecules (*Linda Mgbechidinma et al., 2022*), considering the existence of twenty different amino acids that can be used to derivatize them.

Here, by introducing random mutations along the gene sequence of OmpA using DNA shuffling (*Meyer, Ellefson & Ellington, 2014*), we aimed to increase OmpA's ability to reduce IFT in crude-water systems and identify mutations that may have improved surfactant activity for each of the obtained mutant proteins.

## MATERIALS AND METHODS

### Strains and growth conditions

*E. coli* K-12 W3110/pCA24N OmpA$^+$ was obtained from the ASKA library (*Kitagawa et al., 2005*). It was grown overnight at 37 °C on Lysogeny Broth (LB) agar plates containing 50 μg/mL chloramphenicol. The strain was grown from single colonies in 50 mL of LB medium supplemented with 50 μg/mL chloramphenicol and incubated overnight at 37 °C at 250 rpm.

### DNA shuffling

The *ompA* gene was amplified using external primers (Table 1) that annealed 150 base pairs upstream and downstream of the target gene (Fig. S1). The reaction contained: 1 μl pCA24N OmpA$^+$ template (102 ng), primers EF and ER (0.3 μM each), 5 μl 10X Buffer (Mg$^{2+}$ Plus), 4 μl dNTP Mixture (2.5 mM each), and 5 U of Takara Taq (Takara Bio., Shiga, Japan), for a total volume of 50 μl. PCR using iCycler (Bio-Rad, Hercules, CA, USA) was performed at 94 °C for 4 min, with 35 cycles of 94 °C for 45 s; 55 °C for 30 s; 72 °C for 90 s, and a final extension of 72 °C for ten minutes. 2 μg of the resultant band (1,439 bp) was fragmented randomly with DNase I. The fragmentation was performed in a 50 μl reaction mixture containing 5 μl of 10X buffer (1 M Tris-HCl pH 7.4 and 200 mM MnCl$_2$) and 0.05 U of DNase I. The reaction was incubated for 30 s at 80 °C before the incubation was terminated after 12 min. Following electrophoresis on 1% agarose, fragments between 400 bp and 1,000 bp were purified according to the protocol by the PureLink$^®$ Quick

**Table 1  Primers used in this research. (Supporting Material –Primer Blast Section).**

|  | Name | Sequence | Fragment expected |
|---|---|---|---|
| **External and internal primers** | EF | GTGAGCGGATAACAATTTCACAC | 1,439 bp |
| | ER | GGGCATGGCACTCTTGAA | |
| | IF | AGGCCTATGCGGCCATGAAA | 1,215 bp |
| | IR | AATTGGGACAACTCCAGTGAAAAGT | |
| **Sequencing** | M13 F | GTAAAACGACGGCCAG | |
| | M13 R | CAGGAAACAGCTATGAC | |

Gel Extraction Kit (Invitrogen, Carlsbad, CA, USA). A mixture with 40 µl of purified fragments, 5 ul of 10X Buffer ($Mg^{+2}$ Plus), 2 µl of dNTPs, and 5 U of Takara Taq (Takara Bio., Shiga, Japan) was prepared. PCR was performed at 96 °C for 90 s, with 35 cycles of 94 °C for 30 s; 65 °C for 90 s; 62 °C for 90 s; 59 °C for 90 s; 56 °C for 90 s; 53 °C for 90 s; 50 °C for 90 s; 47 °C for 90 s; 72 °C for 4 min, and a final extension of 72 °C for 7 min.

Using 200 ng from the resulting PCR product, 5 µl 10X Buffer ($Mg^{2+}$ Plus), 4 µl dNTP Mixture (2.5 mM each), 0.15 µl of each primer IF and IR (Table 1), 5 U of Takara Taq (Takara Bio., Shiga, Japan) and 3 µl of DMSO in a total volume of 50 µl, to achieve punctual mutations by mismatched base-pairing. Final PCR with inner primers (IF and IR) is conducted at 94 °C for 30 s; 25 cycles of 98 °C for 10 s; 65 °C for 30 s; 72 °C for 150 s, and final elongation at 72 °C for 7 min. The two randomized resulting bands (1,100 bp and 300 bp) were cloned into pCR4-TOPO® and transferred to competent *E. coli* DH5α. 50 µg/ml ampicillin was used to select clones for library construction.

## Molecular techniques

Plasmid DNA was extracted utilizing the Spin Miniprep Kit (QIAGEN, Inc., Valencia, CA, USA). DNA fragments were isolated from agarose gels using PureLink® Quick Gel Extraction Kit (Invitrogen, Waltham, MA, USA). The recombination reaction was carried out at 22 °C using TOPO TA® Cloning® Kit (Invitrogen, Waltham, MA, USA). In this study, the NanoDrop® ND-1000 Thermo Scientific (Waltham, MA, USA) was used to quantify DNA.

## Screening of biosurfactant activity using 96-well plates

210 µl of LB medium with ampicillin (50 µg/ml) were added to 96-microwell plates, inoculating 10 µl of previously grown bacterial culture for each clone. The plates were incubated at 37 °C overnight. After 20 min of centrifugation at 4,500 rpm, the plate was analyzed using a paper backing sheet with a grid. According to *Walter, Syldatk & Hausmann (2010)*, the clones presenting distorted-looking wells were selected as biosurfactant producers. Tween 20 and *P. aeruginosa* rhamnolipid were utilized as positive controls (*Sajna et al., 2015*).

## Interfacial tension measurement

For each clone and the purified proteins, 5 ml of LB medium was inoculated with single colonies previously grown in LB agar plates with ampicillin (50 µg/ml). When

**Table 2  Characterization for crude-oil used in this research.**

|  | Units | Value |
|---|---|---|
| Density | g/cm$^3$ | 0.95 |
| Density | ° API | 16.3 |
| pH | NA | 6.7 |
| Viscosity @25° C (s$^{-1}$) | cP | 3200 |
| Water | (%v/v) | 6.5 |
| Aromatics | % | 40.21 |
| Resins | % | 21.32 |
| Asphaltenes |  | 8.65 |

OD$_{600}$ reached 0.6, IPTG (isopropyl-b-D-thiogalactopyranoside) was added to a final concentration of 0.5 mM, and incubation at 250 rpm was continued for 4 h at 37 °C for induction of protein expression. Cells were harvested by centrifugation at 4 °C and 3,500 rpm for 15 min. Cells were lysed in a Sonicator VC-750 (SONICS, Newtons, CT, USA) as follows: amplitude 20% and with the pulse set to 1 s × 1 s for 2 min and then kept on ice for 10 min (three times). The supernatant was recovered after centrifugation at 4 °C and 13,000 rpm for 15 min. Each protein extract was added to 2 ml of diluted crude oil to a final concentration of 0.055% (v/v) and homogenized *via* sonication at 19% amplitude and pulse set to 1 s × 1 s for 2 min. The light phase was prepared by diluting crude oil (12° API) with dodecane to a final concentration of 75% (v/v) and mixed using Dissolver DISPERMAT® LC (VMA-GETZMANN GMBH, Hilden, Germany) at 2,500 rpm for 45 min. Interfacial tension was quantified using the Attension Theta Tensiometer (Biolin Scientific, Gothenburg, Sweden). The technique selected was the optical method of the inverted pendant drop, with the drop profile fitting method of Young-Laplace (*Berry et al., 2015*). The image scan was obtained at a rate of 20 frames/s for 5 min. Table 2 lists the properties of the crude oil used in this work.

## DNA sequencing

Inserts were amplified by combining 2.5 µl of 10X Buffer, 1.5 µl of MgCl, 4 µl of dNTPs, 1 µl of each M13 F and M13 R primers (Table 1), and 5 U of Takara Taq (Takara Bio., Shiga, Japan), to a total reaction volume of 25 µl. The reaction took place using the thermocycler iCycler (Bio-Rad, Hercules, CA, USA) at 95 °C for 4 min, 25 cycles of 95 °C for 30 s; 54 °C for 30 s; 72 °C for 90 s; and final elongation of 72 °C for 5 min. Fragments were sequenced at the Universidad de los Andes by Sanger sequencing method. Alignment of all the sequences against OmpA was done using Clustal Omega (*McWilliam et al., 2013*).

## 3D structure prediction and protein analysis

The 3D structures of each protein were acquired from ColabFold v1.5.3: AlphaFold2 using MMseqs2 (*Mirdita et al., 2022*). The top-ranked model structure underwent relaxation using Amber. msa_mode was configured as mmseqs2_uniref_env, and the pair_mode was set to unpaired_paired. num_recycles setting was adjusted to 24, while the remaining options were kept at their default settings. The stability index (*Guruprasad, Reddy & Pandit,*

*1990*) and solubility values for the nine proteins were predicted using the Protparam tool (*Gasteiger et al., 2005*) and Protein-Sol (*Hebditch et al., 2017*) servers, respectively.

## Protein purification by immobilized metal affinity chromatography

First, clone 6 and 12 sequences were synthesized individually in plasmid puc57 in Shanghai Shine Gene[R] Molecular Biotech, then they were digested and ligated in pET6xHN-N (Fig. S2). For each selected clone, 50 ml of LB medium was inoculated with 100 µg/ml of ampicillin from single colonies previously grown on LB agar plates, and allowed to incubate overnight. With 100 mL of LB containing an initial OD of 0.2, a new culture was started for every mutation. When the OD600 reached 0.6, induction with isopropyl-β-D-thiogalactopyranoside (IPTG) at a concentration of 1.5 mM was initiated, and incubation was continued for 5 h at 37 °C. Cells were collected by centrifugation at 4 °C and 4,000 rpm for 20 min. Cell lysis was performed with a tip sonicator as follows: amplitude 37%, for 15 min with pulses of 20 s on and 40 s off. The supernatant was collected after centrifuging at 4 °C and 4,000 rpm for 20 min. The supernatant was purified with cobalt columns (Cobalt chelating resin, 1 mL spin Column, G-Biosciences, St. Louis, MO, USA). The column was equilibrated with Cobalt equilibration buffer (50 mM $NaH_2PO_4$, 300 mM NaCl, 10 mM imidazole pH 7.4), and then the supernatant of the mutations was added to separate columns. Cobalt buffer was added once more, and three washes were collected. To detach the target protein from the resin, heparin wash buffer (50 mM $Na_2HPO_4$, 300 mM NaCl, 250 mM imidazole, pH 8.0) was utilized. For each mutation, four elusions were collected. The purification of mutations was confirmed by tricine-sodium dodecyl sulfate-polyacrylamide gel electrophoresis (tricine-SDS-PAGE) (*Bio-Rad Labs, 2022*). The collected fractions were mixed with sample buffer and denatured for 5 min at 70 °C. Electrophoresis was run for 65 min at a voltage of 140V. The gel was left in a fixing solution (glutaraldehyde 5%), stained with Coomassie blue, and the protein bands were visualized by adding a staining solution.

## RESULTS AND DISCUSSION

### DNA Shuffling for *ompA*

To accomplish DNA shuffling (23), the strain *E. coli* K-12 W3110/pCA24N OmpA[+] was grown on LB agar plates with 50 µg/ml chloramphenicol, and purification of the plasmid pCA24N was carried out (*Winstanley & Rapley, 2000*). Amplification of the *ompA* gene was successfully achieved using EF and ER as external primers (Table 1, Fig. S4). Random fragmentation was conducted by evaluating six different time reactions 2 min, 30 s, 60 s, 90 s, 10 s, 5 s to analyze the fragments' sizes after different times of incubation. Furthermore, the reassembly reaction without primers was carried out with fragments between 400 bp and 1,000 bp. As expected, the resulting fragments obtained are larger than the original gene (Figs. S5 and S6). Final amplification was carried out with dimethylsulfoxide (DMSO) and external primers IF and IR primers (Table 1). Results showed two different fragments, one with a size similar to the original gene, and the other with approximately 400 bp. We previously showed that the biosurfactant performance is possibly associated with its size, mostly affecting the formation of micelles (*Segura et al., 2014a*). Cultures with noticeable

growth were twice plated on LB agar to maintain the concentration of antibiotics in the medium and prevent the formation of satellite colonies. The continued growth of 25 clones was detected, and the library was constructed in glycerol stocks and stored at −80 °C.

## Initial screening of the IFT ability of the engineered OmpA proteins

The 96-well plates with the supernatants of each clone were observed through a grid. The main goal of this test is to identify the grid distortion when viewed through an aqueous medium containing a surfactant. The results of these tests are mainly qualitative and a distortion of the grid for all clones is observed (Fig. S7), showing that all clones still conserved IFT activity. Then, IFT was also evaluated in the Attension® Theta Tensiometer.

## Alignment of sequences and comparison with OmpA

Among the clones studied, nine of them were selected according to the obtained IFT values and consequently sequenced based on the Sanger method on the fact that they exhibited three behaviors: lower, equal, and higher IFT compared to the parental OmpA protein. To identify the protein mutations, we used Clustal Omega EMBOSS Needle (*Huang et al., 2014*). The pairwise sequence alignments show different parts of the original sequence that are conserved in the fragments obtained and some other parts that are deleted around the positions 350 to 420 (of the original sequence) are lost possibly due to the presence of DMSO, conditions of amplification and long time exposure to DNAse. These results show that random mutations are present throughout the original gene (Supporting Material-Pair-wise Sequence Alignments). Departing from the Kyte-Dolittle plots and molecular dynamics to calculate the free energy change when inserting molecules in water-dodecane interfaces, our group previously designed surfactants derived from OmpA (*Aguilera-Segura et al., 2016*). However, we did not find similarities between those sequences and the ones derived from the DNA shuffling so both approaches could provide new structures with enhanced functionality.

## Analysis of isoelectric point and hydropathic properties of the shuffled *ompA* proteins

Here we found that clones 6 and 12 significantly cloaser and lower the IFT compared to OmpA (the negative control (water/crude oil) displayed a value of 25 mN/m). Protein-IFT correlation can be influenced by several factors that permit to explain protein functionality (*Singh et al., 2020*), like repulsive interactions (steric or electrostatic) that can play a big role at the interface. Kyte-Doolittle plots have been constantly used to determine hydrophobicity regions (*Huang et al., 2014*) therefore these plots were obtained for all peptides (Fig. 2).

On one hand clone 12 shows defined hydrophilic-hydrophobic regions with neutral and positively charged zones (Fig. 3), typical for a surfactant when facilitating the interaction at the interface. Moreover, a clear charged-like behavior of the protein is elucidated based on isoelectric point calculation assuming a neutral pH of the sample (Tables 3 and 4). On the other hand, clone 6 does not display a well-defined protein structure (Fig. 4). Nevertheless, the isoelectric point could justify that the protein is charged at neutral pH conditions, possibly explaining the property of the protein to reach a lower IFT value compared to OmpA.

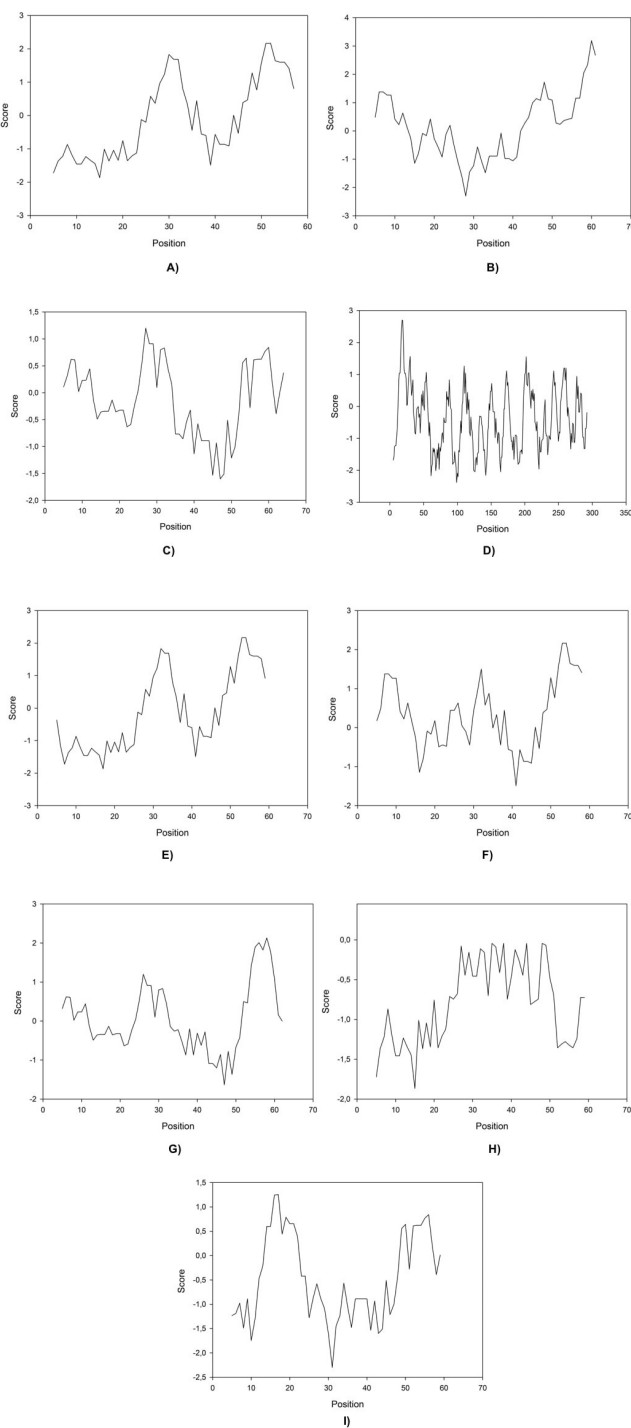

**Figure 2** Kite & Doolittle plot for proteins 1 (A), 5 (B), 6 (C), 7 (D), 12 (E), 13 (F), 16 (G), 21 (H), and 23 (I).

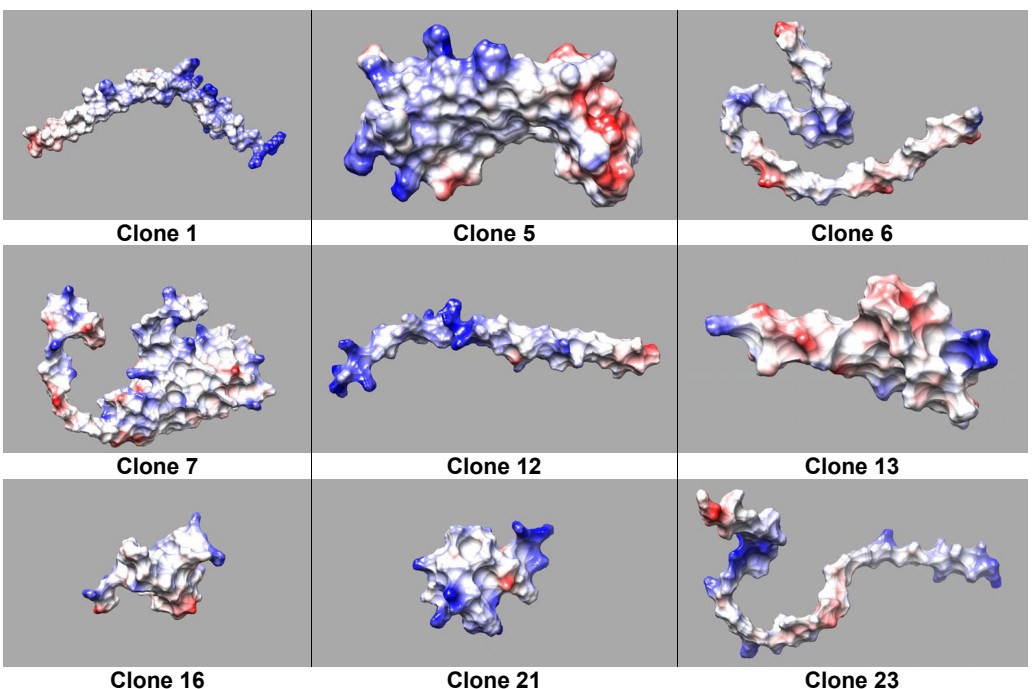

**Figure 3** **Coulombic surface of protein structures predicted by AlphaFold2, calculated using Chimera USCF software.** Red (negatively charged), blue (positively charged), and white (neutral charge).

**Table 3** **Values of IFT reached by every clone, measured with Attension Theta Tensiometer 2 (± represents standard deviation for three biological replicas).**

| Clone | IFT (mN/s) |
|---|---|
| 1 | 27.34 ± 1.23 |
| 5 | 29.69 ± 2.82 |
| 6 | 17.1 ± 3.02 |
| 7 | 36.26 ± 2.47 |
| 12 | 7.22 ± 2.03 |
| 13 | 30.22 ± 3.83 |
| 16 | 23.2 ± 2.03 |
| 21 | 30.02 ± 5.02 |
| 23 | 25.33 ± 2.10 |
| Purified 6 | 9.10 ± 3.02 |
| Purified 12 | 7.24 ± 2.19 |
| OmpA | 14.98 ± 3.01 |
| Tween 20 | 22 ± 0.34 |

Molecular size and weight have been reported to be indicators for surfactant performance because they can explain how easily a molecule could migrate to the interface, without taking into account specific non-bonding (*Bak & Podgorska, 2016*). Our results indicate that size and weight play a minor role in improving IFT activity as there is no direct correlation

**Table 4** Sequences obtained for each protein after DNA shuffling and isoelectric points, as calculated by *Aguilera-Segura et al. (2016)*.

| Protein | Sequence | Isoelectric point |
|---|---|---|
| 1 | KRRPVNCNTTHYRANIRPRIRPWAWHSILCEIVIRSQGRIRLNLQDSLGLILSLASWSLFP | 11.71 |
| 5 | DGQIVIRLTIGRIEFSGREFALADNNFTQNSRVPCPRANSFKPAGLVPLVRVNSELGVIMVIAVS | 9.49 |
| 6 | QETAMTMITPSSELTLTKGTSPAGLNEFALVSGQFHTSRVPCPRANSRPLNSIRPIVSRITIHWPSFY | 10.03 |
| 7 | FHQQQWPDALGHDWALVLLVVTSREFLCEIVMSYDWLGRMPYFGTVENQVLGLRCIQSSGRST DRTGLPNHRPGHLQIRPPWHLCEIVKSNVYGKNHDTGANRYRVKLLPINIGDAHTIGTRPAPKDNTWYTGAK LGWSQYKPEGQAALGFEMGYDWLGRMPYKGPDNGMLSLGVSYRFGQGEATKSNVYGKNHDTGVSPVFAGGVE YAITPEIATRLEYQWTNNIGDAHTIGTRPDNGMLSLGVSYRFGQGEAAPVVAPAPAPAPEVQTKHF TLKSDVLFNFNKATLKPEGQAAL | 8.37 |
| 12 | LVKRRPVNCNTTHYRANIRPRIRPWAWHSILCEIVIRSQGRIRLNLQDSLGLILSLASWSLLP | 11.71 |
| 13 | NDGQIVIRLTIGRIEFSGREFALGHGTLEIQCEIVIRSQGRIRLNLQDSLGLILSLASWSLF | 6.77 |
| 16 | ETAMTMITPSSELTLTKGTSPAGLNEFALVSGQFHTEFIQECHAQGRIRGRIQFALVVLQFTGRRF | 8.33 |
| 21 | KRRPVNCNTTHYRANIRPRIRPWAWHSSVKLLSAHKGEFVTCRTSPFSEGFAWRNHGHSCFLR | 11.49 |
| 23 | KQLPLRQAQNPSLKGLVLQVTNSPLADNNFTQNSRVPCPRANSRPLNSIRPIVSRITIHWPSF | 12.01 |
| OmpA | APKDNTWYTGAKLGWSQYHDTGFINNNGPTHENQLGAGAFGGYQVNPYVGFEMGYDWLGRMP YKGSVENGAYKAQGVQLTAKLGYPITDDLDIYTRLGGMVWRADTKSNVYGKNHDTGVSPVFAGGVEYAITPEI ATRLEYQWTNNIGDAHTIGTRPDNGMLSLGVSYRFGQGEAAPVVAPAPAPAPEVQTKHFTLKSDVLFNFNKATLK PEGQAALDQLYSQLSNLDPKDGSVVVLGYTDRIGSDAYNQGLSERRAQSVVDYLISKGIPADKISARGMGESN PVTGNTCDNVKQRAALIDCLAPDRRVEIEVKGIKDVVTQPQA | 5.6 |

between this variable an the reached IFT. Moreover the isoelectric point for other proteins such as those derived from clones 5, 7, 13, and 16 present approximately neutral values and higher IFT values than OmpA (Table 3), probably corroborating the charge as one of the main factors regarding interface activity in protein design and indicating that proteins with an isoelectric point closer to the solution's pH would take longer to travel to the interface (Table 4).

**Functionality-protein structure relation**

The tertiary structure of each protein was predicted using Alphafold2 (Fig. 2). Moreover we obtained the distribution of charges for clones (Fig. 4). It can be noticed that the protein produced by Clone 12 has positive charged residues, while the protein produced by Clone 13 has both (positive and negative) charges in its structure. Studies conducted by *Nurdin (2014)*, show that surfactants with both charges can interact with aromatic compounds naturally present in crude oil, obstructing the adequate organization of the surfactant at the oil-water interface. The behavior in the migration of ionic proteins found in this research, can be attributed to these electrostatic interactions with components present in the oil. Possibly indicating that proteins that are able to reduce the interfacial tension rapidly migrate to the interface, are mainly positively charged.

A deeper examination of the 3D structures shows that alpha-helixes were predicted for clone 12 (Fig. 4). *Dexter, Malcolm & Middelberg (2006)* designed a peptide with such a secondary structure and the capacity to stabilize emulsions and foams based on forming cohesive interfacial films. Therefore, the role of alpha-helix secondary structures is documented and could explain the mechanism of the protein expressed by clone 12 for decreasing IFT. On the other hand clone 6 displayed a non-typical structure, possibly

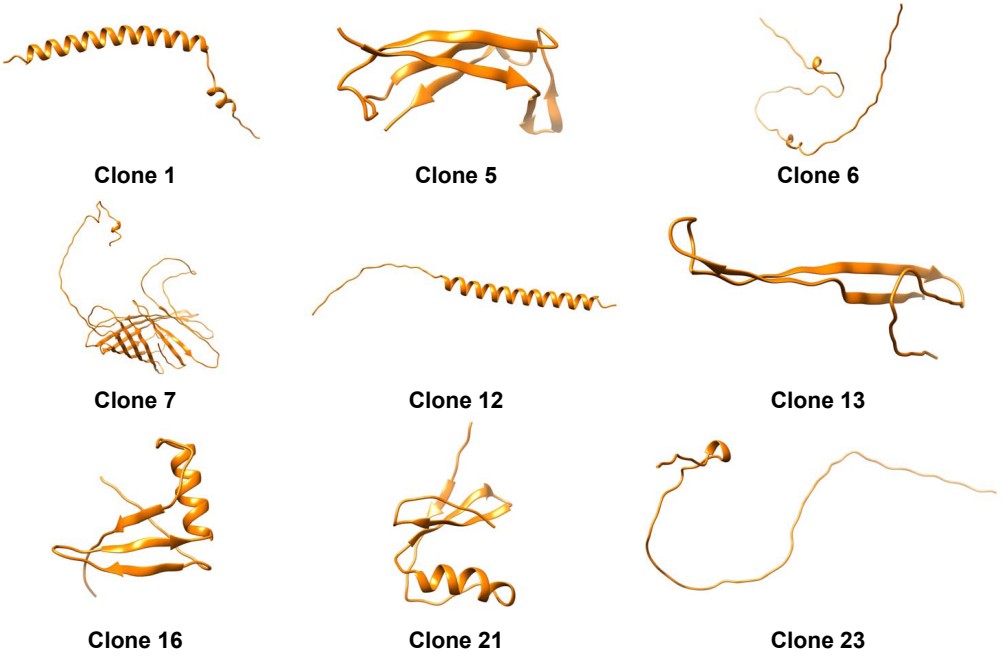

**Figure 4** Three-dimensional predicted structures from AlphaFold2 for clones 1, 5, 6, 7, 12, 13, 16, 21, and 23.

indicating that conformational changes could take place during the protein migration to the interface which structure prediction algorithms are not able to explain (*Fillery-Travis, Mills & Wilde, 2000*).

We calculated the stability index (*Gasteiger et al., 2005*) and predicted solubility (*Hebditch et al., 2017*) values for the nine clones (Table 5). A protein with a stability index less than 40 is predicted to be stable, while a value above 40 suggests potential instability. These results indicate that clones 6 and 12 are unstable proteins and also exhibit higher solubility compared to the other clones. Plotting the IFT values and stability index (Fig. 5) reveals an apparent inversely proportional relationship, indicating that a more unstable protein has a higher surfactant capacity. When a protein is more unstable, it is more likely to undergo structural changes. Based on our findings, it is conceivable that a more unstable protein may harbor more flexible regions, enabling it to better accommodate interfaces. This, in turn, facilitates its capacity to diminish surface tension and function as a surfactant. The optical density at 280 nm was measured to assess the stability of the clones and OmpA, finding a constant absorbance value. This method is crucial for determining the concentration of protein in the samples, as the absorbance at this specific wavelength is indicative of the presence of aromatic amino acids, primarily tryptophan and tyrosine, which absorb UV light. The constant absorbance reading suggests that the protein concentration remained stable over the course of the experiments, indicating no

**Table 5  Stability index (SI), interfacial tension (IFT) and solubility for each protein.**

| Clone | SI | Solubility |
| --- | --- | --- |
| 1 | 79.94 | 0.543 |
| 5 | 22.51 | 0.646 |
| 6 | 50.72 | 0.56 |
| 7 | 30.72 | 0.297 |
| 12 | 77.26 | 0.606 |
| 13 | 40.91 | 0.47 |
| 16 | 49.08 | 0.377 |
| 21 | 66.92 | 0.575 |
| 23 | 47.18 | 0.686 |

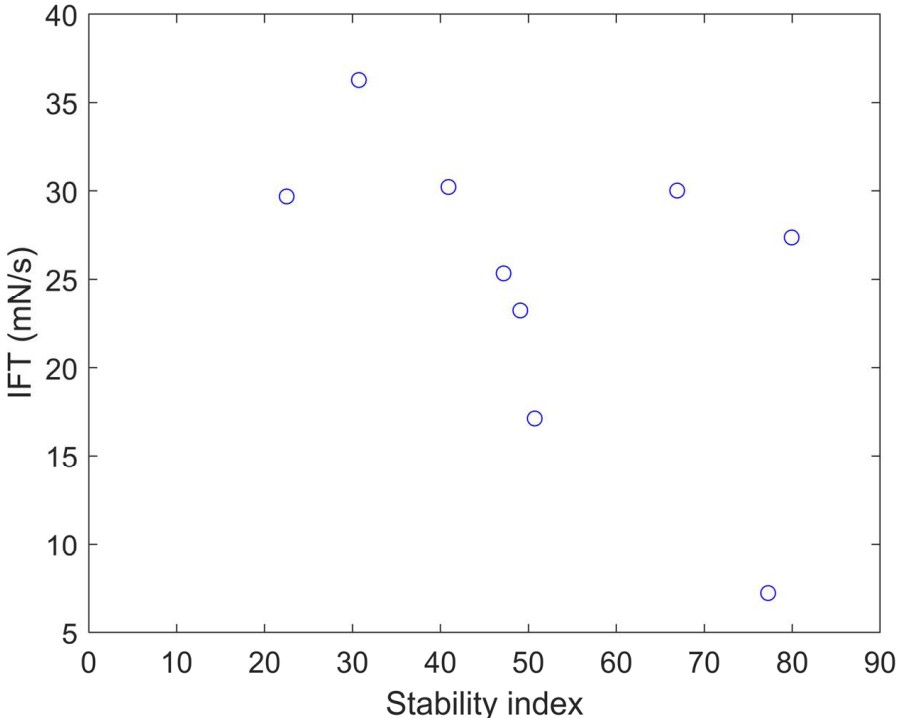

**Figure 5  Stability index *vs.* IFT plot for clones 1, 5, 6, 7, 12, 13, 16, 21, and 23.**

significant degradation or loss of the clones and OmpA, thus providing a reliable measure of their stability.

## IFT evaluation for purified 6 and 12 clones proteins

For the measurement of the IFT at the dodecane-water interface, first, the samples obtained in the purification were quantified, obtaining 0.588 mg/mL for elution with mutant 6 and 0.451 mg/mL for mutant 12. Then, we found that the IFT obtained for the purified clones was lower compared to their correspondent protein extract and lower than OmpA corroborating the individual effect of the individual proteins (Table 3).

## CONCLUSIONS

In this work, we demonstrated that DNA shuffling can be useful for improving biosurfactants IFT activity at crude oil and water interfaces, departing from a transmembrane protein that was earlier demonstrated to be useful for EOR. Based on the analysis of the sequence of mutants, we found that charge is possibly one of the crucial features for explaining the performance of each molecule. Several clones derived from OmpA displayed improved functionality reflected in the lower IFT values compared to the original protein (OmpA). For the same purposes, rational (molecular simulations) and random approaches (DNA shuffling) would provide different candidates that could replace existing pollutant polymers currently utilized in the market for the same purposes. After purification, both mutants still had a lower IFT value than the original protein (OmpA). Using Tween 20, a higher interfacial tension result was obtained compared to the mutants. Then, it was shown that it is possible to create and obtain ecologically acceptable biosurfactants that have a greater effect on reducing the IFT compared to a conventional surfactant.

### Funding
The authors received no funding for this work.

### Competing Interests
The authors declare there are no competing interests.

### Author Contributions

- Vanessa Lucía Nuñez Velez conceived and designed the experiments, performed the experiments, analyzed the data, prepared figures and/or tables, authored or reviewed drafts of the article, and approved the final draft.
- Liseth Daniela Villamizar Gomez performed the experiments, analyzed the data, prepared figures and/or tables, authored or reviewed drafts of the article, and approved the final draft.
- Jhon E. Mendoza Ospina performed the experiments, analyzed the data, prepared figures and/or tables, authored or reviewed drafts of the article, and approved the final draft.
- Yasser Hayek-Orduz performed the experiments, analyzed the data, prepared figures and/or tables, and approved the final draft.
- Miguel Fernandez-Niño conceived and designed the experiments, analyzed the data, authored or reviewed drafts of the article, and approved the final draft.
- Silvia Restrepo Restrepo conceived and designed the experiments, analyzed the data, authored or reviewed drafts of the article, and approved the final draft.
- Óscar Alberto Álvarez Solano analyzed the data, authored or reviewed drafts of the article, and approved the final draft.
- Luis H. Reyes Barrios conceived and designed the experiments, analyzed the data, authored or reviewed drafts of the article, and approved the final draft.

- Andres F. Gonzalez Barrios conceived and designed the experiments, analyzed the data, authored or reviewed drafts of the article, and approved the final draft.

## DNA Deposition

The following information was supplied regarding the deposition of DNA sequences:

GenBank OR536314

Seq2 OR536315

Seq3 OR536316

Seq4 OR536317

Seq5 OR536318

Seq6 OR536319

Seq7 OR536320

Seq8 OR536321

Seq9 OR536322

## Data Availability

The raw measurements are available in the Supplemental File.

## Supplemental Information

Supplemental information for this article can be found online at http://dx.doi.org/10.7717/peerj.17239#supplemental-information.

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
