# Peer review of "DNA shuffling to improve crude-water interfacial activity in biosurfactants with OmpA protein of Escherichia coli"

_PeerJ, doi:10.7717/peerj.17239_

## Round 0.1 · original submission · Major Revisions

Dear authors,
After the review by two experts in the field, there are some points that need clarification and some corrections are needed. Based on the two experts' reports I think the work is interesting and the data supports the findings. Please address point by point all the concerns of the reviewers. I also kindly request to include the number of replicates done on each experiment and include either SEM or statistical analysis. Also, in Table 4 it is desirable to estimate the stability index of each clone obtained here and predicted solubility. Also, please include two structural models for the clones that are showing disordered structure as supplementary material.
Thank you for choosing PeerJ.
All the best for your research moving forward.
Bernardo

·

Basic reporting

It is OK, well written and clear, the references are adequate as well as the figures and tables.

Experimental design

In general it is OK; however, in table 3 “Values of IFT reached by every clone, measured with Attension Theta Tensiometer”.
The measurements of the IFT per clone lack a dispersion value such as standard deviation or SEM.
Does this mean the measurements were done in only one culture per strain? Or that the values of dispersion are omitted.
It is important to clarify and if the measurements were done once repeat at least 2 times more, provide the dispersion and then apply statistical test to determine if the differences of IFT found are significant.

Validity of the findings

that depend on the answer to the question asked in the section 2.

Additional comments

Abstract: “colitrans” add a space.
L. 47 “is a surface-active protein due to its ability to form biofilms” the bacteria form the biofilm, not the protein by itself…please rephrase.
L. 88 “The cells were centrifuged for 5 minutes at 13,000 rpm.” For what? What was do next?
L. 159 “the sequence of clone 6 and 12 clones” do you mean the sequences of the ompA genes present in the clones 6 and 12? please rephrase.
L. 170, L. 173 add a space between number and units eg. “300mM”
L. 263 “faculty” better “property”
L. 209 “protein..” remove one “.”

Reviewer 2 ·

Basic reporting

The article is well written, although it is necessary to review minor errors. The references are sufficient and appropriate for the foundation of the work, and the figures are appropriately used. The introduction is adequate to state the question being addressed correctly. It is hoped that in the final version, the clarity of the figures will be of better resolution. Likewise, it is necessary to clarify the work's title, directly indicating that it is a DNA shuffling of the ompA gene. There are some minor typographical errors; for example, on the presentation page, it says Escherichia colitransmembrane, and it is necessary to define the abbreviations when they are used; for instance (AlnA) is not specified when it is used for the first time in line 45

Experimental design

The methods are described correctly. However, many candidates of little interest are handled instead of focusing the work mainly on those that show the desired surfactant properties.

Validity of the findings

A question that remains is whether or not the achieved recombinant proteins maintain their biodegradable property. At the end of their results in lines 271 to 275, it is stated that the IFT found from the purified clones was lower than the OmpA protein, which requires clarification since the IFT of OmpA is 14.98, that of clone 6. It is 17.1, and 12 is 7.22, so they show lower and higher values among their clones. Please clarify this point. On the other hand, it would be desirable to have a diagram that allows us to appreciate the changes in the clones compared to the original protein; there are various graphic models for this.

Additional comments

In general, the work presents an orderly exploration of the generation of mutants to obtain ecologically friendly surfactant proteins, and its conclusions are sufficient to base its methodological approach to the problem.

---

## Round 0.2 · Minor Revisions

Dear authors,

Both reviewers have delivered their reports and indicate that the manuscript has attended all the issues detected. However, Reviewer 2 indicates that there is a single experiment that is crucial for the manuscript, to demonstrate that the clonal products obtained in this work degrade at the same rate as the original protein, this must be demonstrated experimentally in order to provide a full view of the protein derivatives. The stability indices are not enough to support this claim. I kindly request to include the data in the paper.

Thank you for the important manuscript you provided.

All the best,
Bernardo

·

Basic reporting

The authors addressed all my comments and I think their work is now suitable for publication.

Experimental design

The authors addressed all my comments and I think their work is now suitable for publication.

Validity of the findings

The authors addressed all my comments and I think their work is now suitable for publication.

Additional comments

The authors addressed all my comments and I think their work is now suitable for publication.

Reviewer 2 ·

Basic reporting

The authors attended to the points of the first review, so I have no further comments.

Experimental design

The stability indices are added to the work, which is why what was previously stated is fulfilled.

Validity of the findings

Although the authors are confident in the biodegradable properties of the clonal products, this should be demonstrated.

Additional comments

No comment

---

## Round 0.3 · accepted · Accept

Dear authors,

Thank you so much for providing the additional data that supports the theoretical observations of the generated peptides. Based on the reports of the reviewers and the overall assessment of the manuscript, I am happy to let you know that the paper is now suitable for publication. Thank you for choosing PeerJ.

Congratulations and all the best for your research moving forward.
Best regards,

Bernardo